# HER2 Low Breast Cancer: A New Subtype or a Trojan for Cytotoxic Drug Delivery?

**DOI:** 10.3390/ijms24098206

**Published:** 2023-05-04

**Authors:** Marina Popović, Tajana Silovski, Marija Križić, Natalija Dedić Plavetić

**Affiliations:** 1Department of Oncology, University Hospital Centre Zagreb, 10000 Zagreb, Croatia; tsilovsk@kbc-zagreb.hr (T.S.);; 2School of Medicine, University of Zagreb, 10000 Zagreb, Croatia

**Keywords:** HER2 low, HER2 overexpression, *HER2* amplification, HER2 testing, antibody drug conjugate, trastuzumab deruxtecan, T-DXd, HER2 heterogeneity, HER2-enriched

## Abstract

Despite the great progress made in the understanding of the biological behavior of certain types of invasive breast cancer, there is still no single histological or molecular classification that encompasses such diversity and accurately predicts the clinical course of distinct breast cancer subtypes. The long-lasting classification of breast cancer as HER2-positive vs. HER2-negative has recently come into question with the discovery of new antibody drug conjugates (ADC), which are proven to be remarkably efficient in treating HER2-low breast cancer. The HER2-low paradigm has challenged the traditional understanding of HER2 overexpression and emphasized the need for more robust HER2 testing in order to encompass HER2 intratumoral heterogeneity and spatial distribution more accurately. It is yet to be seen if low HER2 will remain merely a marker of HER2-equipped tumors targetable with ADCs or if distinctive molecular and phenotypic groups within HER2-low tumors will eventually be discerned.

## 1. HER2 Signaling Pathway

Breast cancer (BC) is a heterogeneous disease that includes different BC subtypes with distinguishable histological and molecular characteristics, clinical presentations, and responses to treatment and prognosis. Conventionally, BC classification has relied mostly on characteristics such as tumor morphology, estrogen and progesterone hormone receptor (HR) expression, human epidermal growth factor receptor 2 (HER2) expression, and proliferation marker Ki67. More recently, other important prognostic and predictive factors, such as androgen receptor expression, response to neoadjuvant therapy, tumor infiltrating lymphocyte and programmed death-ligand 1 (PD-L1) expression, subtypes of triple negative BC, gene mutations and the so-called “integrated cluster” groups, were also included in the WHO classification of BC [1].

Since gene profiling was not available for routine clinical use, in order to assess the outcome of the disease more accurately and guide the treatment decisions, it was decided to group breast tumors into so-called surrogate intrinsic groups, defined via routine histology and immunohistochemistry (IHC) [2,3]. Based on IHC staining of three protein markers, estrogen receptor (ER), progesterone receptor (PR) and HER2 receptor, complemented by in situ hybridization (ISH), at the St. Gallen Breast Cancer Conference in 2013, BC surrogates intrinsic groups were consensually defined as luminal A—ER+, PR ≥ 20%, HER2− and Ki67 < 20%; luminal B HER2 negative—ER+, HER2−, PR < 20% and/or Ki67 > 20%; luminal B HER2 positive—ER+, HER2+; HER2 positive—HER2+; ER−, PR− and triple negative—ER−, PR−, HER2− [4]. Although not without differences, classification based on IHC and molecular classification showed acceptable concordance.

Approximately 15% of patients with BC overexpress HER2, with the age-adjusted rate of HR+HER2+ BC being 12.9, and that of HR-HER2+ being 5.2 new cases per 100,000 women, based on 2015–2019 cases. Among HER2-negative BC, the age-adjusted rate of HR+ was 87.4 and that of HR- was 13.2 new cases per 100,000 women, respectively [5]. According to current clinical practice guidelines, all newly diagnosed and recurrent BC need to be tested for HER2 overexpression [2,6].

The transmembrane glycoprotein HER2/ErbB2 is a member of the ErbB family of protein kinases coded by the HER2/ERBB2 gene located on chromosome 17q21. Other members of the family include HER1/ErbB1/EGFR, HER3/ErbB3 and HER4/ErbB4 receptors. There are structural and functional differences between individual receptor family members. HER2 is the only member of the ErbB family that has no known ligands [7], and the HER3 receptor lacks the functional intracellular thyrosin kinase domain [8]. HER1, HER3, and HER4 selectively bind different ligands including neuregulins, epidermal growth factor (EGF), transforming growth factor-α (TGF-α), betacellulin, epiregulin, heparin-binding EGF-like growth factor (HB-EGF) and amphiregulin [9]. In normal conditions, ligand binding is obligatory for signal transduction. HER2 signalization can therefore only be activated via heterodimerization with other ErbB family receptors or via homodimerization in a ligand-independent manner following HER2 overexpression [10]. The formation of HER1, HER2, and HER4 homodimers results in weak signal transduction, but HER2 overexpression promotes the formation of HER2/HER1, HER3 or HER4 heterodimers that lead to more potent and prolonged signaling [9]. HER2 acts as an oncogene in BC. Overexpression of HER2 leads to the constitutive activation of different EGFR signaling pathways, i.e., the PI3K/AKT/mTOR and RAS-RAF-MEK-ERK pathways, which when overly activated lead to oncogenic transformation and tumor progression [11].

## 2. HER2 Testing—From Protein Overexpression to Copy Number Gain

HER2 testing is carried by both IHC and ISH. IHC is performed to quantify HER2 protein expression and HER2-positive BC is defined by strong and complete IHC membrane staining of more than 10% of cells (3+). In the case of an equivocal finding (2+), defined as invasive BC with weak to moderate complete membrane staining observed in 10% of tumor cells, single- or dual-probe ISH is conducted to evaluate HER2 gene amplification. If the HER2 IHC score is 2+ and gene amplification is present, the tumor is categorized as HER2-positive. In all other cases (HER2 2+ without HER2 gene amplification, HER2 1+ or HER2-0), the tumor is considered HER2-negative.

From a pathologist’s point of view, one of the main challenges is distinguishing HER2 1+ from HER2 2+ as accurately as possible, given that IHC 1+ will be regarded as HER2-negative, whereas IHC 2+ will be followed by reflex ISH testing, leading to a re-classification of the tumor as HER2-positive in approximately 25% of cases [12]. This can be particularly difficult when evaluating samples with unusual HER2 expression that do not meet the criteria for IHC 2+ but are found to be ISH-amplified. This can be a rare case of micropapillary or gland-forming carcinomas showing moderate to intense but incomplete basolateral or U-shaped staining, or intense circumferential IHC staining in ≤10% of tumor cells. Both cases should be considered as IHC 2+-equivocal and reflexed to ISH [13,14]. Cases of negative or 1+ HER2 staining with ISH-proven gene amplification are exceedingly rare [15], whereas HER2 2+ membranous staining is considered a bad predictor of HER2 amplification [15,16,17].

ISH is based on a quantitative measurement of the number of copies of the HER2 gene using a single- or dual-probe assay. A single probe assay enumerates copies of the HER2 gene per nucleus only, while a dual-probe assay also includes a probe for the centromeric region of chromosome 17 (CEP17), using a ratio of HER2 gene signals to chromosome 17 copy number control signals to determine gene amplification [18]. The American Society of Clinical Oncology/College of American Pathologists (ASCO/CAP) preferentially recommends the use of dual-probe instead of single-probe ISH assays. Single-probe ISH amplification testing is considered positive when there is an average HER2 copy number of ≥6.0 signals/cell and <4.0 signals/cell is regarded as negative. For an average HER2 copy number that is ≥4.0 but <6.0, dual-probe testing is required. Dual-probe ISH results are divided according to the HER2/CEP17 ratio, with the cut-off point set at ≥2. More rigorous interpretation criteria are suggested for the less common patterns seen in about 5% of all cases, i.e., a HER2/CEP17 ratio of ≥ 2 and an average HER2 copy number of <4.0 signals/cell or a HER2/CEP17 ratio of < 2 and an average HER2 copy number of ≥4.0 signals/cell [13].

Overexpression of HER2 in BC occurs most commonly through HER2 gene amplification and is associated with a more aggressive phenotype [19]. Although HER2 overexpression in the absence of a high HER2 copy number is generally uncommon, ranging from around 3%, as reported by Pauletti et al. [20], to 11%, shown by Hoang et al., using HercepTest [17], some type of transcriptional or translational events could be in the background of such an event. The difference in the research methods, i.e., the type of antibodies and scoring system used could explain the percentage discordance seen in individual studies.

## 3. HER2-Low—A New Clinical Entity

Following phase I clinical studies that demonstrated efficacy of the new antibody drug conjugates in the setting of tumors with weak HER2 protein expression [21,22], Tarantino and colleagues proposed a new entity—HER2-low BC. By definition, tumors with a HER2 IHC score of 1+ or 2+ without amplification upon routine ISH are classified as HER2-low [23]. HER2-low tumors lack HER2 amplification and are therefore regarded not as HER2-driven but rather HER2-equipped [14]. The proportion of HER2-low tumors to all breast cancer cases is considerable, with just over halve of the HER2-negative patients being HER2-low [23,24]. The majority (68%) of HER2-low tumors are HER2 1+, regardless of HR status [24]. The rate of HER2-low tumors is significantly higher in HR-positive disease (up to 65%) than it is in triple-negative BC (TNBC) (around 35%) [24,25,26].

Clarifying whether or not HER2-low expression translates into better treatment outcomes and overall survival rates, in comparison to those seen in HER2-0 BC, has become one of the burning fields of research. If this is true, it would imply that HER2-low is a distinct BC subtype, requiring a distinct treatment approach. Several trials have addressed this matter, but yielded somewhat contradictory results, in metastatic BC (mBC) [27,28,29,30], early BC in general [31,32,33,34], and even in HR+ early BC [35,36]. The observed discrepancies could be at least in part due to the retrospective multi-institutional research designs, lack of central HER2 validation, lack of prior treatment information, etc. Meanwhile, studies looking at early TNBC are more consistent, showing no impact of a HER2-low phenotype on disease-free or overall survival [37,38].

In a pooled analysis of four prospective neoadjuvant clinical trials that included 1098 patients with HER2-low and HER2-0 BC treated with combination chemotherapy, patients with HER2-low tumors had a significantly lower pathological complete response (pCR) rate than those with HER2-0 tumors did overall, including the HR+ subgroup. This was not seen in the HR- subgroup. On the other hand, HER2-low BC patients had a significantly longer survival than those with HER2-0 BC did. This was also true for the HR- subgroup, while in the HR+ subgroup no survival benefit was observed. The authors postulated that HER2-low BC should be regarded as a specific BC subtype [39]. Disputing this assumption are the results from another large trial using data from all consecutive 5235 patients with early HER2-low BC undergoing surgery at Dana-Farber Brigham Cancer Center. The authors found no statistically significant differences in the pCR rate or in overall survival between HER2-low and HER2-0 tumors, regardless of if the patient had HR+ or TNBC [39]. Keeping in mind the considerable difference in the percentage of HR-driven tumors between HER2-low and HER2-0 BC, favoring the former, elucidating the true effect of HER2 low expression on the clinical course and outcomes could prove to be rather challenging, as nicely pointed out by Nicolo et al. [40].

## 4. Transcriptional Subtypes—PAM50 within HER2-Low BC

A turning point in the understanding of the biology of BC took place in 2000 when Perou and colleagues, by measuring complementary DNA (cDNA), recognized similar patterns in gene expression between different BC samples and were able to identify four distinctive groups based on characteristic “molecular portraits”: (1) the ER+/luminal group in which the expression of genes characteristic of luminal epithelial breast cell prevailed, (2) the basal group with the expression of genes typical of myoepithelial and basal breast cells and in which there is characteristically no expression of a hormone or HER2 receptor, (3) the HER2-enriched (HER2-E) group with a high expression of ERBB2 genes and (4) a group with similar-to-normal breast tissue [41]. At least two distinct subgroups were subsequently defined within the luminal group: luminal A and luminal B—marked by a high expression of genes responsible for mitosis and proliferation [42].

Numerous studies have confirmed the existence of intrinsic groups using different gene signatures, with partial agreement in tumor categorization depending on the type of microarray used, the threshold for gene expression determination and the level of heterogeneity within the tumor. The highest concordance was in the basal category, and the lowest was in luminal B and HER2-E tumors. The PAM50 gene expression test was developed as a standard method for categorizing BC into intrinsic molecular groups. PAM50 classifies tumors as luminal A, luminal B, HER2-E and basal group [43].

Analyzing PAM50 and individual gene expression data from 1320 HER2-low patients, Schettini and colleagues found a lack of enrichment of the HER2-E subtype within HER2-low disease. HER2-low tumors were characterized by a high expression of luminal-related genes, while no significant subtype distribution was observed in TNBC, with a basal-like subtype (85%) prevailing, leading to the belief that HR-positive HER2-low tumors are more distinct biological entities, in contrast to HER2-low TNBC tumors [24].

## 5. HER2 Amplification Is Independent of the Subtype

The analysis of the genomic profiles of 3155 BC samples from the TCGA (The Cancer Genome Atlas) database, METABRIC (Molecular Taxonomy of BC International Consortium) and the USO1062 study showed that HER2 amplification exists in all PAM50 groups. When assessing the genomic profile of HER2-amplified tumors across all subtypes, the authors found that the genomic events in HER2-amplified tumors resembled those of the underlying subtype, in lieu of being characteristic of those driven by HER2 amplification. Only PIK2CA mutations, GATA3 mutations and BRCA1 deletions were associated with HER2 amplification in an individual subtype. No signals indicating that HER2 amplification is a distinct transcriptional subtype were found. The authors concluded that HER2 amplification is merely an event on top of a luminal or basal transcriptional and mutational state [44].

## 6. HER2-Low Is an “Old Friend” Already Seen in Pivotal Trastuzumab Clinical Trials

Preclinical trials conducted over two decades ago indicated that trastuzumab was indeed most effective in cells with HER2 overexpression and amplification. The history of anti-HER2 agents began back in 1998 with the FDA approval of trastuzumab as a monotherapy for metastatic HER2-positive BC after progression to other lines of therapy. After trastuzumab was proven to be effective in the later lines of treatment, Vogel et al. conducted a trial of first-line trastuzumab given as a monotherapy to patients with HER2 2+ or 3+ metastatic BC (mBC) [45]. The objective response rate seen with trastuzumab in the ITT population was 26%, with responses being seen only in patients with HER2 3+ tumors, and no response being seen in tumors that were categorized as HER2 2+. According to the ISH-determined HER2 amplification in 108 assessable patients, the response rate was 34% in those with HER2-amplified tumors and 7% in those without amplification.

That the combination of trastuzumab and chemotherapy as first-line treatment yields better results than chemotherapy alone does was shown by Slamon and colleagues in a trial that included 469 patients with HER2 2+ and 3+ mBC. The addition of trastuzumab to chemotherapy resulted in a longer time to disease progression (median: 7.4 vs. 4.6 months), and longer survival (median survival: 25.1 vs. 20.3 months) [46]. HER2 2+ patients were well represented in the study, ranging from 20 to 30 percent, depending on the study arm. Testing for amplification was not mandatory for the enrollment of HER2 2+ patients. The efficacy of adding trastuzumab to chemotherapy according to the IHC HER2 status was not reported in the initial publication.

The influence of HER2 amplification, as determined by ISH, on clinical outcomes in women with HER2 2+ or 3+ mBC treated with trastuzumab was analyzed in three clinical trials [47]. The HER2:CEP17 ratio of ≥2 was regarded as positive. The proportion of ISH-positive patients was comparable in all three trials, with 78% of patients having HER2-amplified tumors overall. Clinical benefit from trastuzumab therapy appeared to be restricted to patients with HER2-amplified mBC in all observed studies. 

The lessons from the metastatic setting of trastuzumab efficacy in the patients with IHC HER2 2+ tumors were learned and the ground was set for adjuvant trials.

## 7. HER2-Low BC in Early Adjuvant Trials

The two large trials, the NSABP trial B31 (hereafter B31) and the NCCTG N9831 (hereafter N9831) trial, compared conventional a doxorubicin and cyclophosphamide followed by paclitaxel (AC-T) chemotherapy regimen with or without trastuzumab being administered concomitantly or being administered after paclitaxel is. The trials were amended for a joint analysis comparing the trastuzumab arm with the control. Trastuzumab therapy was associated with a 33% likelihood of death reduction, and trastuzumab combined with paclitaxel after AC improved survival due to the sustained reduction in cancer recurrence among women with HER2-positive BC [48,49]. Central testing of archived tumors specimens from B-31 and N9831 trials revealed that the portion of tested samples was HER2-negative as found by both IHC and ISH using a central assay. However, these patients derived a similar degree of benefit, with a similar HR for DFS (0.51) to that of patients with HER2-positive disease [50,51]. The finding indicated that, as opposed to what was concluded in the metastatic setting, in the adjuvant setting even tumors with lower HER2 expression and/or amplification could benefit from the addition of trastuzumab to standard chemotherapy treatment. The proposed explanation was that in the adjuvant setting, the immune system is less compromised and therefore a lower level of HER2 expression is needed to trigger the antibody-dependent cellular cytotoxic (ADCC) response [52,53].

Based on these retrospective observations, the NSABP B-47 (hereafter B-47) trial was designed to determine whether or not the addition of trastuzumab to chemotherapy improves invasive disease-free survival (IDFS) in women with resected node-positive or high-risk node-negative BC reported as HER2 1+ or 2+ and ISH-negative (hereafter, HER2-low) [54]. B-47 did not confirm the hypothesis. A lack of benefit was evident in all patient subsets, and administration of HER2-directed therapies in adjuvant and neoadjuvant settings remained limited to populations identified as HER2-positive according to the ASCO/CAP guidelines updated in 2018.

## 8. Antibody–Drug Conjugates—One Drug to Treat Them All

The cornerstone of antibody drug conjugates (ADC) was set as early as in 20th century with Paul Erlich´s receptor theory that gave rise to the “magic bullet concept” [55], essentially indicating target-guided drug delivery. By the end of 2022, 13 different ADCs were granted FDA approval, with only two of them having HER2 as a target.

Trastuzumab–emtansine (T-DM1) was the first FDA-approved antibody—drug conjugate (ADC) in breast cancer treatment. T-DM1 is composed of a potent tubulin polymerization inhibitor called DM1, which is covalently linked to trastuzumab via a non-cleavable thioether linker. T-DM1 carries an average of 3.5 DM1 molecules per one molecule of trastuzumab [56,57]. Following several positive trials, the FDA approved T-DM1 for patients with advanced BC [58,59] and those with residual disease after neoadjuvant therapy [60]. However, T-DM1 demonstrated poor activity for early BC with low and/or heterogeneous HER2 expression in the neoadjuvant setting [61], further reinforcing the view that only HER2 overexpressing tumors are HER2 pathway-dependent and therefore benefit from HER2-directed therapy.

Trastuzumab–deruxtecan (T-DXd), a new generation ADC, was the second to be approved by the FDA, after the remarkable response rate and progression-free survival shown in patients with heavily pretreated HER2-positive metastatic breast cancer in the DESTINY-Breast01 trial [62]. In 2022, based on the results of the DESTINY-Breast03 trial [63], T-DXd was also approved as a second-line treatment for patients with unresectable or metastatic HER2-positive breast cancer who have received a prior anti-HER2-based regimen either in the metastatic setting or in the neoadjuvant or adjuvant setting and have developed disease recurrence during or within six months of completing therapy.

T-DXd is a conjugate of the humanized HER2-targeted antibody that has the same structure as trastuzumab, and DXd, a potent topoisomerase I inhibitor. The two are linked via an enzymatically cleavable peptide and an amino-methylene moiety, which reduce hydrophobicity and provide stability in systemic circulation. As nicely put by the drug discovery researchers, T-DXd was specifically designed to improve the characteristics of other anti-HER2 ADCs. It has a payload of 10-fold higher potency than irinotecan does, a high antibody–drug ratio of almost 8, a stable linker payload, a payload with a short systemic half-life, a cleavable linker and the bystander tumor effect [64]. The bystander tumor effect explains the effect of the drug on the neighboring cells that do not express HER2, after interstitial linker cleavage and the release of the membrane-permeable free payload (Figure 1) [65]. This allows the problem of heterogeneity in HER2 expression, one of the great challenges in cancer treatment, to be overcome.

A kind of paradigm shift in the way HER2 expression is apprehended, beyond HER2-positive BC defined per “standard of practice” guidelines, happened in 2022 when the results of the DESTINY-Breast04 study were published [66]. The DESTINY-Breast04 study was a phase 3 trial involving patients with HER2 low unresectable BC, or mBC. Patients were stratified according to a HER2 low status (IHC1+ vs. IHC2+, ISH-negative), HR status (positive vs. negative), prior CDK 4/6 (cyclin-dependent kinases 4/6) inhibitor therapy and lines of chemotherapy for metastatic disease (1 vs. 2 lines). A little more than half of the patients were HER2 IHC1+. All patients treated with T-DXd had a significantly longer PFS as well as OS, compared to patients on the treatment of the physician’s choice (TPC) (∆4.8, 9.9 vs. 5.1 months, 23.4 vs. 16.8, and 23.9 vs. 17.5 months, respectively). Interestingly, PFS and OS improvement was also observed in rge hormone receptor-negative group who received T-DXd vs. the group with the TPC (∆5.6, 8.5 vs. 2.9 months, and ∆9.9, 18.2 vs. 8.3 months, respectively). The benefit of T-DXd was consistent regardless of whether or not the patients were previously treated with a CDK 4/6 inhibitor or had received more than one line of chemotherapy in the metastatic setting. Objective response rate and clinical benefit rate were high both in HR-positive and -negative patient groups, bearing in mind that the number of HR-negative patients was relatively low (N = 58). In line with DESTINY-Breast04 results, T-DXd was approved as the first HER2-directed therapy for patients with HER2-low mBC. Additionally, just like that, the field of HER2-low shifted from being one with a testing flaw to becoming an attractive new area of research (Table 1).

## 9. New ADCs with Proven Activity in HER2 Low BC

Trastuzumab–duocarmazine (SYD 985) is a conjugate of the anti-HER2 antibody bound to a DNA alkylating duocarmycin preload with a low drug to antibody ratio (DAR) of 2.8:1 [67]. After the improvement in PFS seen with trastuzumab–duocarmazine in the phase 3 Tulip trial [68] which included patients with HER2-positive mBC, trastuzumab–duocarmazine has also shown clinical activity in heavily pretreated HER2-low mBC patients [21].

The innovative anti-HER2 ADC disitamab–vedotin (RC48–ADC), composed of a novel anti-HER2 humanized antibody hertuzumab linked to the microtubule inhibitor monomethyl auristatin E (MMAE) through a cleavable linker, revealed activity in the cohort of 48 HER2-low mBC patients, with an ORR of 39.6% and a mPFS of 5.7 months [69]. A phase III trial investigating the efficacy and safety of RC48–ADC in HER2-low BC is ongoing (NCT04400695).

MRG002, a novel anti-HER2 ADC composed of modified trastuzumab conjugated through a cleavable linker to the MMAE payload, was investigated in a phase 2 trial which included pretreated HER2-low mBC patients. Both HER2 1+ and HER2 2+/ISH- BC patients obtained similar ORRs, disease control rates and mPFSs (33%, 75%, and 5.6 months, respectively) [70].

## 10. Overcoming Resistance to ADCs

Although ADCs have made a revolutionary change in the treatment of mBC, patients still experience disease progression due to evolving mechanisms of drug resistance. Abelman RO et al. categorized mechanisms of resistance according to whether they are related to changes in antigen expression, the processing of the ADCs, or the chemotherapy payload [71]. Caveolae-mediated endocytosis and loss of HER2 expression are associated with resistance to T-DM1 [72,73]. The latter was successfully overcome with the discovery of T-DXd, which exhibits efficacy in both HER2-low and HER2-0 tumors due to the bystander effect [74]. However, the complex structure of the ADCs makes them susceptible to versatile drug resistance mechanisms. Decreased penetration into the cell by barriers and abnormal endosomal transit [75], as well as payload-related resistance, are just some of the mechanisms behind the reduction in drug effectiveness that will have to be addressed either by new generations of ADCs, combination therapies, or other yet to be recognized alternatives.

A new field of research that has already yielded promising results is the combination of ADCs with immunotherapy. The rationale behind this comes from preclinical trials indicating that the addition of ADCs could enhance the effectiveness of immune checkpoint inhibitors (ICIs) through tumor neoantigen formation, the activation of immune cell-like dendritic cells, and an increase in PD-L1 expression [76,77,78]. One of the trials showing promising early safety and efficacy results is the phase 1b/2 BEGONIA trial, investigating the combination of durvalumab and T-DXd, in HER2-low TNBC patients [79]. The combination of T-DXd and nivolumab in HER2-expressing BC patients was investigated in a phase 1b trial, but the HER2-low subgroup was too small to draw any conclusions [80]. The results of trials investigating combination of T-DXd with pembrolizumab are still awaited (NCT04042701).

## 11. HER2 Intratumoral Heterogeneity—No Longer a Pitfall but a Treatment Possibility?

As a tumor progresses, an increasing number of mutations accumulate, which do not have to be the same in all tumor cells, in a process called clonal evolution. They contribute to tumor heterogeneity and make treatment more difficult. In contrast with the genetic cause of intratumoral heterogeneity, the cancer stem cell (CSC) hypothesis postulates that intratumoral heterogeneity may also arise due to a distinct cell transition under specific environmental pressure, as in post transcriptional modification during cell differentiation. The CSC hypothesis cannot fully explain non-genetic tumor variability, and other mechanisms that underlie cellular phenotypes, such as the epigenetic landscape, must also be taken into consideration [81].

The genomic heterogeneity of HER2-positive tumors has long been recognized, and back in 2008 the ASCO/CAP group agreed to convene an expert panel to address the issue of HER2 amplification with intratumoral heterogeneity, giving rise to discrepant results between IHC and ISH assays. The panel defined HER2 genetic heterogeneity as the presence of more than 5% but less than 50% of infiltrating tumor cells with a ratio higher than 2.2 [82]. In 2013, ASCO/CAP guidelines referred to intratumoral heterogeneity as the presence of distinct cell populations within the tumor, with discrete aggregates (clusters) of amplified cells comprising >10% of the total tumor population being considered significant [83]. The two other possibilities of HER2 intratumoral heterogeneity are the so-called intermixed heterogeneity, as in the diffuse intermingling of HER2-amplified and -non-amplified cells, and isolated heterogeneity, both of which are more frequently encountered in HER2 2+ tumors and are generally scored as amplification-negative. The prevalence of HER2 intratumoral heterogeneity was estimated to be as high as ≈30%, depending on the given study [84,85,86,87], and was a common finding in HER2-equivocal tumors [84,88].

There are several proposed mechanisms underlying the expression of HER2 in the absence of HER2 amplification in HER2-low tumors, one being the crosstalk between the HER2 receptor network and the ER genomic and non-genomic mechanisms of action promoting HER2 overexpression, especially under conditions of endocrine treatment pressure [14]. A good example is the upregulation of nuclear receptor co-activator mRNA and HER2 during treatment with aromatase inhibitors, leading to increased HER2 mRNA levels in tumors originally not overexpressing HER2 [89]. Other mechanisms include HER2 upregulation through activation of the NF-kB pathway after chemotherapy and radiation [90] and epigenetic changes leading to increased HER2 expression [23]. Back in 2013, Ithimakin and colleagues conducted an experiment administering trastuzumab with or without docetaxel to HER2-amplified or luminal tumor xenografts early in the course of the disease as well as in the advanced stage of the disease, after the tumor was established. In HER2-amplified cells, trastuzumab showed a significant effect on tumor growth regardless of the setting in which it was applied, whilst in luminal tumors with no HER2 copy gain, the beneficial effect of trastuzumab was only shown when administered in the early (adjuvant) setting. What was even more interesting was that when analyzing the expression of HER2 in aldehyde dehydrogenase (ALDH)-positive cells (ALDH taken as a marker of CSC), more than 90% of ALDH-positive cells displayed HER2 expression, while only 40% of ALDH-negative cells expressed HER2 in primary luminal tumors. In contrast, in HER2-amplified tumors, there was no significant difference in HER2 expression in ALDH+ versus ALDH−cells. It should also be noted that in the tumor-invasive front, in HER2-amplified tumors, HER2 expression was more uniform [26].

When reflecting on tumor heterogeneity, discordance between HER2 expression in the primary tumor and regional or distant metastases, upon metastatic diagnosis, as well as upon disease progression, must be acknowledged. A large study comparing immunophenotypic tumor profiles between matched breast cancer primaries and metastatic sites from the Epidemio-Strategy-Medical-Economical (ESME)-MBC database, showed a change rate between primary tumor and metastases, at metastatic diagnosis, for the HER2 status to be 7.8%, with an absence of overexpression/amplification in 45.2% of cases and gain in 54.8% of cases. After the first progression, the HER2 modification rate was 10% with a loss of HER2 overexpression in 50.9% of cases and a gain in 49.1% of cases. [91]. For HER2-low tumors, the discordance rates were much higher, proving that HER2-low is highly unstable during disease progression, as reported by Miglietta et al. [25]. In the cohort of 547 BC patients with paired samples of primary tumors and metastatic sites, HER2-low breast cancer accounted for 45% of the entire HER2-negative cohort, with the metastatic cohort being enriched for HER2 low tumors compared to the primary breast cancer population. The overall rate of HER2 discordance between the primary tumor and the metastatic site was 38%, mostly represented by a HER2-0 status switching to a HER2-low status (15%) and a HER2-low status switching to a HER2-0 status (14%). Among HER2-positive breast cancer patients showing HER2 loss from primary to recurrent breast cancer, 77% maintained some level of HER2 expression, exhibiting a HER2-low phenotype. Again, association between HR status and HER2-low expression was observed. Importantly, HER2 expression upon recurrence was not affected by the type of sample, reconfiming the reliability of the relapse BC phenotype irrespectively of the biopsied anatomical site.

## 12. Improving the Robustness of HER2 Testing

The FDA approved several IHC anti-HER2 antibodies and single or dual ISH DNA probe tests for HER2 overexpression and HER2 amplification evaluation [92]. In 2022, the FDA expanded the indications of the approval of the PATHWAY anti-Her2/neu (4B5) rabbit monoclonal primary antibody (Ventana Medical Systems, Inc.) to include HER2 protein testing for patients with BC who may benefit from treatment with T-DXd [93].

Despite the numerous available and approved tests, both IHC and ISH have downsides, and many argue that HER2 testing lacks very much needed robustness. IHC is a semi-quantitative method, and interpretation of IHC results can sometimes be challenging even for an experienced pathologist. Methodology limitations such as improper specimen handling, artifacts, a lack of proper validation tests and analytical testing failures can lead to false test results causing improper patient management [94]. At the same time, factors including intratumoral HER2 amplification heterogeneity and CEP17 copy number gain or loss can contribute to an inaccurate HER2 amplification assessment. CEP17 gain in the absence of HER2 amplification is associated with HER2-negative tumors having no clinical benefits with anti-HER-2 treatment [95]. Nevertheless, ISH CEP17 copy number gain is frequently observed in BCs. It may arise due to true chromosome 17 polysomy, or due to an amplification of the centromeric or pericentromeric portion of chromosome 17 [18,96,97]. Earlier research indicated that true chromosome 17 polysomy is a rare event [18,98,99] but a more recent translational study carried out by Halilovic and colleagues showed that CEP17 copy number gain results as a consequence of widespread aneuploidy with the gain of multiple chromosomes, including chromosome 17 [100]. Furthermore, data suggest that CEP17 copy number may have prognostic value in reflecting intratumoral chromosomal instability, which is associated with worse outcomes in BC patients [101].

Molecular techniques based on the quantitative evaluation of HER2 messenger RNA (mRNA), such as reverse transcription quantitative PCR (RT-qPCR), have been suggested as a better alternative to the combination of IHC and ISH methods for HER2 testing [102,103,104].

In light of HER2-low BC, and moreover, the proven efficacy of treatment with the new ADC outside of HER2-positive tumors defined as such by current testing guidelines, new methods of HER2 testing using gene protein assays and next-generation sequencing (NGS) may help to overcome the problem of HER2 heterogeneity, using single-slide HER2 assessment to delineate patients who will benefit from anti-HER2 treatment [105]. As shown by Zhang and colleagues, looking at 774 BC patients that had their HER2 status determined by IHC, ISH and targeted genomic profiling of cancer related genes, at the cut-off of a copy number of 2.62, NGS could identify a IHC/ISH-determined HER2-negative status with 97.8% specificity, and at the cut-off of ≥3.62, it could identify a IHC/ISH-determined HER2-positive status with 99.8% specificity, respectively. Patients classified by NGS as having lowly amplified tumors, i.e., a copy number between 2.62 and 3.62, had heterogeneous IHC/ISH results and a distinct mutational profile, indicating that this is a more precise way for selecting patients for ADC treatment [106].

## 13. Conclusions

In light of the development of new ADCs and the DESTINY-Breast trials confirming the remarkable efficacy of T-DXd in both HER2-overexpressed and HER2-low tumors, the long-lasting classification of BC as HER2-positive vs. HER2-negative has become obsolete. Until new data are available that help discern more precisely the distinctive molecular and phenotypic groups within HER2-low tumors and the implication they might have on disease outcome and therapeutic approach, for clinical purposes, HER2 expression should be considered part of a growing body of targets for the delivery of a highly potent payload. Improving the robustness of HER2 testing by using new methods such as RT-qPCR, NGS, multiple(x) protein analysis or even AI (artificial intelligence)-mediated solutions might help to identify more complex cellular phenotypes and minimize the effect of HER2 heterogeneity and spatial distribution on HER2 testing quality, identifying patients who will benefit most from ADC treatment as well as those whose tumors will prove to be resistant to such treatment.

## Figures and Tables

**Figure 1 ijms-24-08206-f001:**
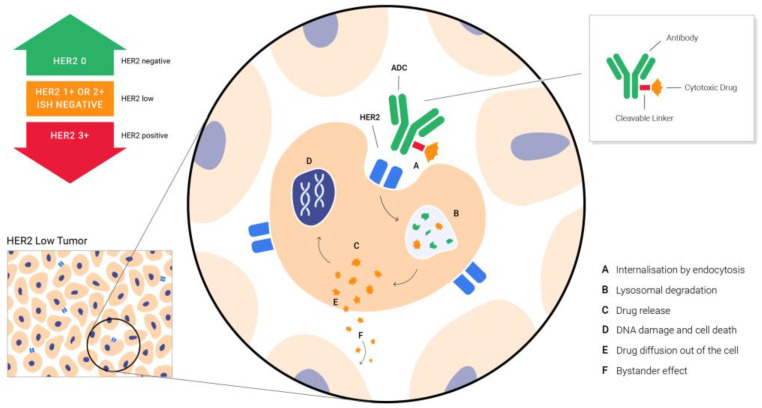
Targeting HER2 low BC with ADCs—the bystander effect that enables the delivery of the drug to the neighboring cells that do not express HER2, after interstitial linker cleavage and release of membrane-permeable free payload.

**Table 1 ijms-24-08206-t001:** Efficacy of HER2-directed therapy in HER2 low patients—from testing flaw to the investigational arm.

Clinical Trial	Phase	N	Setting	Study Population	Treatment in the Investigational Arm/Control Arm	Efficacy of Anti-HER2 Therapy * HER2 Low Patients
Slamon et al., 2001 [46]	3	469	Advanced/metastatic; 2nd- or 3rd-line	HER2 2+ and 3+ BC (IHC)	Trastuzumab + chemotherapy vs. chemotherapy	mOS ITT 25.1 vs. 20.3 months
				* ISH not carried out		* Not reported
Vogel CL et al., 2002 [45]	2	114	Advanced/metastatic; 1st-line	HER2 2+ and 3+ BC (IHC)	Trastuzumab monotherapy, single arm	ORR ITT 26%
				79 patients (ISH+) 29 patients (ISH−)		* ORR HER2 ISH-positive vs. HER2 ISH-negative 34% vs. 7%
Romond EH et al., 2005 [48] NSABP B-31 NCCTG N9831	Both phase 3	3351	Early/adjuvant	HER2 3+ or *HER2*-amplified as determined by ISH	Trastuzumab + chemotherapy vs. chemotherapy	HR for DFS 0.48; *p* < 0.0001 * carried out by Paik S et al., [50] * carried out Perez et al., [51]
Paik S et al., 2008 [50]	Subanalysis	1787	Early/adjuvant	NSABP trial B-31	Trastuzumab + chemotherapy vs. chemotherapy	HER2-positive—HR for DFS 0.47 (0.37–0.62), *p* < 0.001
				174 pts reclassified to HER2-negative		* HER2-negative—HR for DFS 0.34 (0.14–0.80), *p* = 0.014
Perez EA et al., 2010 [51]	Subanalysis	1783	Early/adjuvant	N9831	Trastuzumab + chemotherapy vs. chemotherapy	HER2 positive—HR for DFS 0.46, 0.49, and 0.45 (IHC 3+, HER2/CEP17 ≥ 2.0, both)
				103 pts reclassified as HER2-negative		* HER2 negative—HR for DFS 0.69, 0.54, and 0.51, *p* = 0.26, 0.12, and 0.14 (IHC 0 to 2+, HER2/CEP17 < 2.0, both)
Fehrenbacher L et al., 2020 [54]	3	3270	Early/adjuvant	HER2-low BC	Trastuzumab + chemotherapy vs. chemotherapy	* IDFS 89.8% vs. 89.2% (HR 0.98; 95% CI, 0.76 to 1.25; *p* = 0.85)
Filho OM et al., 2021 [61]	2	164	Early/neoadjuvant	HER2-positive BC	TDM1 + pertuzumab, single arm	* pCR non-heterogeneous vs. heterogeneous
				* 10% pts reclassified as HER2-heterogeneous (ISH)		43.4% vs. 10%
Banerji et al., 2021 [21]	1	99	Advanced/metastatic, heavily pretreated patients	HER2-positive, HER2-low BC	Trastuzumab–duocarmazine; single arm	ORR HER2-positive BC 33% * ORR HER2-low ER+ 28% * ORR HER2-low ER- 40%
Modi S. et al., 2020 [22]	1b	54	Advanced/metastatic heavily pretreated patients	HER2-low BC	Trastuzumab–deruxtecan; single arm	* ORR 37% mDoR (95% CI, 8.8 mo to not evaluable)
Modi S. et al., 2022 [66] DESTINY-Breast04	3	557	Advanced/metastatic ≥2nd-line	HER2-low BC	Trastuzumab–deruxtecan vs. physician’s choice	* mPFS 9.9 vs. 5.1 mo (HR 0.50; *p* < 0.001) * mOS 23.4 vs. 16.8 mo (HR 0.64; *p* = 0.001)

* referring to the HER2 Low Patient group.

## Data Availability

No new data were created or analyzed in this study. Data sharing is not applicable to this article.

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
