# Peer review of "HER2 Low Breast Cancer: A New Subtype or a Trojan for Cytotoxic Drug Delivery?"

_ijms, 2023, doi:10.3390/ijms24098206_

Round 1
Reviewer 1 Report
The manuscript entitled "HER2 low breast cancer: a new subtype or a Trojan for cytotoxic drug delivery?" raises an exciting issue in treating breast cancer with different options depending on the classical and new diagnostic tools.
I found the manuscript is interesting and well-written however, I have some suggestions:
It would be better if the authors described the role of pharmacogenetics in the resistance/response of cancer patients to the available ADC option.
The quality of English is very good.
Author Response
Thank you for taking the necessary time and effort to review our manuscript.
We sincerely appreciate your valuable comment and have included a new section on the mechanisms of resistance to antibody drug conjugates, and the possible ways to overcome them. We have highlighted the change we made within the manuscript.
Reviewer 2 Report
Popović et al. outline in their review the characteristics of HER2-low breast cancer, while also highlighting the treatment options present nowadays for this type of cancer.
I want to congratulate the authors for the thoroughness with which this review has been written. However, I believe there are some flaws that need to be addressed.
- Firstly, the authors do not clearly define HER2-low breast cancer nowhere in the text, as it is the main subject of the review. Some information on it is spread among the text, but a section containing the clear definition of this type of breast cancer is needed.
- Secondly, I believe there is too much information on the HER2-positive breast cancer inserted throughout the text, which adds to the difficulty of following the text. I suggest a re-evaluation of the entire review, in order to minimize the unnecessary information.
- Although well written, the extreme long paragraphs make the text hard to follow. I suggest splitting some of the paragraphs, especially the ones in the section “HER2 low – an “old friend” already seen in pivotal trastuzumab clinical trials “.
- I recommend mentioning the authors of the study described in the “HER2 amplification is independent of the subtype” section.
- I suggest adding some information on new ADC, such as trastuzumab duocarmazine and disitamab and their potential use in trating HER2-low BC.
I recommend reading two papers recently published on the same subject, that may elucidate information needed for this review and better help the authors in making the necessary changes: Nicolo et al. (PMID:36844387) and Zhang et al. (PMID:36612123).
The authors could also include some information about other types of medication used to treat HER2-low BC, such as checkpoint inhibitors.
I hope my suggestions will help with the improvement of the manuscript.
Author Response
We trully appreciate the time and effort you put into writing this meticulous review of our paper and the suggestions you made to help improve its content and presentation.
We have highlighted the changes within the manuscript. Here is a point-by-point response to the risen comments and concerns.
1. Firstly, the authors do not clearly define HER2-low breast cancer nowhere in the text, as it is the main subject of the review. Some information on it is spread among the text, but a section containing the clear definition of this type of breast cancer is needed.
Response: We included a new section in the manuscript clearly defining HER2 low breast cancer, as suggested.
2. Secondly, I believe there is too much information on the HER2-positive breast cancer inserted throughout the text, which adds to the difficulty of following the text. I suggest a re-evaluation of the entire review, in order to minimize the unnecessary information.
Response: We re-evaluated the entire review and tried to eliminate the excess information on the HER2 positive breast cancer focusing more on the HER2 low subtype.
3. Although well written, the extreme long paragraphs make the text hard to follow. I suggest splitting some of the paragraphs, especially the ones in the section “HER2 low – an “old friend” already seen in pivotal trastuzumab clinical trials “.
Response: We reorganized some of the paragraphs, including the paragraph “HER2 low – an “old friend” already seen in pivotal trastuzumab clinical trials “, in order to make them shorter and more comprehensive.
4. I recommend mentioning the authors of the study described in the “HER2 amplification is independent of the subtype” section.
Response: We corrected the reference accordingly.
5. I suggest adding some information on new ADC, such as trastuzumab duocarmazine and disitamab and their potential use in treating HER2-low BC.
Response: We added the information on new ADC and check-point inhibitors and their potential role in treating HER2 low breast cancer.
Thank you for the suggested literature, it was indeed helpful and provided us with new insights into the subject which we included in the manuscript.
Reviewer 3 Report
The paper is interesting and in accord with the literature is quite complete.
Contribution of all authors is significant and could be interesting for scientist.
Analysis and data interpretation are adequacy.
I would suggest :
-supplement the section on immunohistochemical diagnosis of HER2 by pointing out that there are several unusual patterns of immunohistochemical expression of HER2 in breast cancer (Grassini et al DOI: 10.1159/000524227);
-Include a figure illustrating low HER2 expression in breast cancer (this would improve the manuscript);
-Line 69 replace [3+] and [2+] with (3+) and (2+).
Author Response
Thank you for the effort and expertise that you contributed towards reviewing our manuscript.
We are trully grateful for your valuable comment pointig out the unusual patterns of the immunohistochemical expression of HER2 in breast cancer. We have read the suggested article and made changes in the manuscript accordingly.
We included a figure illustrating HER2 low expression as a target for the antibody drug conjugate treatment.
We corrected the wrong bracket in line 69.
The changes were highlighted within the manuscript.
Round 2
Reviewer 2 Report
I would like to congratulate one more time the authors for the thoroughness with which the paper has been written.
The authors performed all the necessary changes. The manuscript is in a good form for publication.
Reviewer 3 Report
The manuscript in this new version is much more complete. The publication of this new version is recommended.